# MA-BBOB: Many-Affine Combinations of BBOB Functions for Evaluating AutoML Approaches in Noiseless Numerical Black-Box Optimization Contexts

**Diederick Vermetten**[1] **Furong Ye**[1] **Thomas Bäck**[1] **Carola Doerr**[2]

[1]Leiden Institute for Advanced Computer Science (LIACS), Leiden University, The Netherlands
[2]Sorbonne Université, CNRS, LIP6, Paris, France

**Abstract**  Extending a recent suggestion to generate new instances for numerical black-box optimization benchmarking by interpolating pairs of the well-established BBOB functions from the COmparing COntinuous Optimizers (COCO) platform, we propose in this work a further generalization that allows multiple affine combinations of the original instances and arbitrarily chosen locations of the global optima.

We demonstrate that the MA-BBOB generator can help fill the instance space, while overall patterns in algorithm performance are preserved. By combining the landscape features of the problems with the performance data, we pose the question of whether these features are as useful for algorithm selection as previous studies have implied.

MA-BBOB is built on the publicly available IOHprofiler platform, which facilitates standardized experimentation routines, provides access to the interactive IOHanalyzer module for performance analysis and visualization, and enables comparisons with the rich and growing data collection available for the (MA-)BBOB functions.

## 1 Introduction

Despite a long tradition of developing automated Machine Learning (AutoML) approaches for numerical black-box optimization contexts [3, 12, 28], empirical evaluations are heavily centered around very few benchmark collections. One of the most popular collections is the BBOB suite [10] of the COmparing COntinuous Optimizers (COCO) platform [9]. The BBOB suite was originally designed to help researchers analyze the behavior of black-numerical black-box algorithms in different optimization contexts. Over time, however, BBOB has been used for many other purposes, including evaluating AutoML methods, even though the problems were never designed to be suitable for this task.

With the increasing popularity of the BBOB benchmarks, wide availability of shared performance data enabled the application of, e.g., algorithm selection methods [12]. To achieve these algorithm selectors, a representation of the problem space is required based on which the performance of different algorithms can be predicted. In the case of BBOB, the most commonly used representation makes use of Exploratory Landscape Analysis (ELA), which has been shown to be able to accurately distinguish between BBOB problems [20, 27].

A key problem of algorithm selection based on BBOB problems lies in the ability to test how well the results generalize. One approach is to use a leave-one-function-out method [23], where the selector is trained on 23 functions and tested on the remaining one. This generally leads to poor performance, as each problem has been specifically designed to have different global function properties. As such, another common method is to leave out a set of problem instances for testing. This way, the selector is trained on all types of problems. However, this has a high potential to overfit the particular biases of the BBOB problems, an often overlooked risk.

To remedy these potential issues, the ability to construct new functions which fill the spaces between existing BBOB functions could be critical. If the instance space can be filled with new problems, these could be used to not only test the generalizability of algorithm selection methods, but also more generally to gain insights into e.g., the relation between the ELA representation of a problem and the behavior of optimization algorithms.

Filling the instance space is a topic of rising interest within the optimization community [1, 19, 22, 34]. While some work has been conducted to create problem instances that reflect the properties of real-world applications or obtain similar characteristics of the existing problems, other work is trying to generate diverse instances. For example, symbolic regression and simulation of Gaussian processes have been applied to generate benchmarks reflecting real-world problem behaviours in [35] and [17, 29]. On the other hand, research in generating diverse instances of combinatorial optimization has been conducted in [4, 5, 16, 19]. Regarding black-box numerical optimization, approaches based on Genetic Programming (GP) have succeeded in generating novel problem instances with controllable characteristics defined by their ELA features in [21], in which the authors used ELA features of BBOB instances as a baseline to regenerate similar instances and design diverse instances. However, to obtain problems with desired characteristics, the GP needs to be executed for each dimension. A recent paper proposed a different perspective on generating new problem instances for numerical optimization. In their paper, Dietrich and Mersmann propose to create new problems through weighted combinations of BBOB problems. By creating these affine combinations of existing problems, it seems that the ELA features can transition smoothly between the two component functions. Moreover, affine combinations of two BBOB problems were applied to analyze the behavior of optimization algorithms in [32]. The paper's results demonstrated that the algorithms' performance alters along the weights of two combined problems.

In this paper, we extend upon the modified version of the affine BBOB combinations [32] by generalizing to combinations between any number of BBOB functions. Through doing this, we address the concerns regarding the scaling of the component functions and the impact of the location of the global optimum. We also propose a modified mechanism to sample weights to avoid potential biases resulting from including too many problems.

From the proposed many-affine problem generation method, we sample 1 000 instances, for which we perform both an ELA based analysis as well as an analysis of the performance of a set of algorithms. By combining these results in a simple algorithm selection model, we raise the question of whether or not the ELA features are sufficiently representative to create a generalizable algorithm selection model.

In summary, our **key contributions and findings** are:

1. We introduce MA-BBOB, a generator of arbitrary affine combinations of the 24 BBOB functions. We explain the rationales behind the various design choices, which include the location of the optimum, the scaling used for interpolating the different functions and the way of sampling in functions from this space. The resulting generator is build on the IOHprofiler platform, which enables equivalent benchmarking setups to the original BBOB problems.

2. We analyze 1 000 randomly sampled instances in $2d$ and in $5d$ via Exploratory Landscape Analysis (ELA [20]) and show that the combined MA-BBOB functions cover the space between the original 'pure' BBOB functions quite well, with the exception of some of problems like the linear slope and ellipsoid problem, which are essentially only available in the 'pure' BBOB functions, but disappear in the MA-BBOB instances with non-trivial weights.

3. We compare the performance of five black-box optimization algorithms on the original BBOB and the 1 000 randomly sampled MA-BBOB instances and show that the rank distribution changes slightly in favour of the CMA-ES algorithms and to the disadvantage of RCobyla.

4. Finally, we also perform per-instance algorithm performance prediction studies on MA-BBOB. The results confirm that the regression accuracy is better when the training set includes generalized BBOB functions. However, we also observe a considerable performance gap between ELA based regression models and those trained with full knowledge of the weights that are used to construct the test instances. These results indicate that the current set of ELA features fail to capture some instance properties that are crucial for algorithm performance, a shortcoming that we expect to motivate future research on the design of features for numerical black-box optimization.

## 2 Background

**The BBOB Problem Suite**. The BBOB collection [10] is one of the main components of the COCO framework [9]. It is heavily used in the black-box optimization community for evaluating derivative-free numerical optimization techniques. On the original BBOB suite of 24 single-objective, noiseless optimization problems [10], hundreds of different optimization algorithms have been tested [2].

One key reason for the popularity of this suite is the ability to create independent instances of the same problem, which are generated by applying transformations in the domain and the objective space. These transformations include rotation, scaling of objective value and moving the location of the global optimum. They allow researchers to evaluate possible bias in their algorithms, and are hence an important component of algorithm benchmarking.

The availability of many instances are also a key enabler for the evaluation of AutoML approaches in black-box optimization contexts. Since not all instances are easily accessible via the original COCO implementation, we have made direct access to the instances available in our IOHprofiler benchmarking environment [7, 33].

**Affine Function Combinations**. While the availability of numerous instances per each BBOB function facilitates AutoML studies, it has been observed that the generalization ability of models trained on BBOB and tested on independent problems is disappointing [13, 15]. This motivated the design of new problems to extend the existing BBOB suite. One such approach was proposed in [8]. It suggests to consider affine combinations of two different problem instances [8]. The resulting problems were analyzed with respect to their fitness landscapes, as seen via exploratory landscape analysis (ELA [20]). They have been shown to smoothly connect their component functions in a reduced-dimensionality ELA space. This seems to imply that we can use these problems to connect any pair of existing problems, which would significantly add to the instance space.

In our follow-up study [32] we recently proposed a modified version of creating these affine function combinations, see Sec. 3.1 for details. We used these functions to compare the performance of five selected black-box optimization algorithms and showed that the behavior differences are not as smooth as the differences in ELA space. In several cases, combinations of two functions are best solved by a different algorithm than the one which solved the component problems.

## 3 The MA-BBOB Benchmark Suite

### 3.1 Scaling of Function Values

When combining multiple functions to create a new benchmark problem, one key factor which impacts the landscape is the scaling of the combined functions. Since we are interested in taking affine combinations of existing functions, a difference in scale might lead one function to dominate all others, leading to limited coverage of the feature space.

The original affine BBOB functions proposed in [8] make use of a tuning procedure for finding useable weights. While this allows for selecting suitable problems, it makes it more challenging to just randomly sample a set of new problems. We therefore suggested an alternative way to generate the affine combinations in [32]. This change is two-fold: each component problem $f$ is first transformed by subtracting the global optimum value $\min f$. This way, we know that each

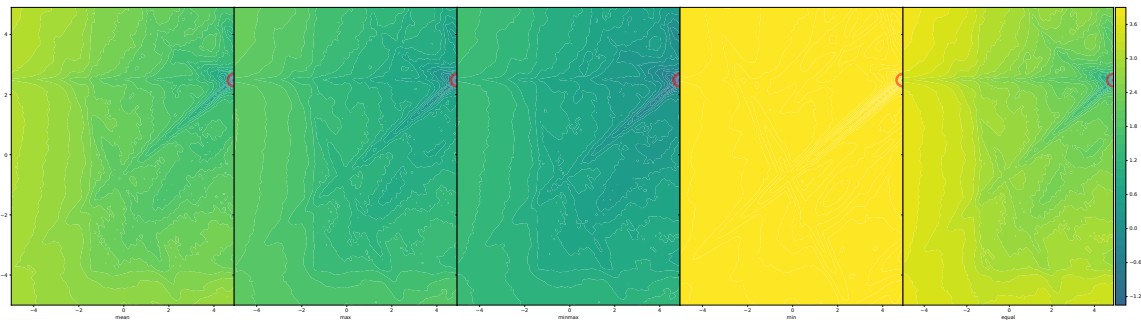

Figure 1: Log-scaled fitness values of an example of a single many-affine function with 5 different ways of scaling. The first 4 are taking the mean, max, $(\max + \min)/2$ and min of 50 000 random samples to create the scale factor, while the 'equal' option does not make use of this scaling.

| Function ID | 1 | 2 | 3 | 4 | 5 | 6 | 7 | 8 | 9 | 10 | 11 | 12 |
|---|---|---|---|---|---|---|---|---|---|---|---|---|
| Scale Factor | 11.0 | 17.5 | 12.3 | 12.6 | 11.5 | 15.3 | 12.1 | 15.3 | 15.2 | 17.4 | 13.4 | 20.4 |
| Function ID | 13 | 14 | 15 | 16 | 17 | 18 | 19 | 20 | 21 | 22 | 23 | 24 |
| Scale Factor | 12.9 | 10.4 | 12.3 | 10.3 | 9.8 | 10.6 | 10.0 | 14.7 | 10.7 | 10.8 | 9.0 | 12.1 |

Table 1: Final scale factors used to generate MA-BBOB problems.

component functions optimum function value is set to 0. Then, instead of arithmetic weighting, a logarithmic combination is used to limit the impact of scale differences. While this simplifies the procedure of generating random function combinations, BBOB functions can sometimes differ by multiple orders of magnitude, which still produces some bias in this procedure.

To address this shortcoming in MA-BBOB, we have investigated different scaling procedures. We still scale the global optima and perform a logarithmic transform, but we now add a normalization step. This transforms the log-precision values into an approximation of $[0, 1]$, and then maps this back to the commonly used BBOB domain $[10^{-8}, 10^2]$. This is achieved by taking the log-transformed precision (capped at $-8$), adding 8 so the minimum is at 0 and dividing by a *scale factor*. The aim of this procedure is to make sure that the target precision of $10^2$ is similarly easy to achieve on all problems.

In order to select appropriate scale factors, we need to determine practical limits of the function value for each BBOB function. We do this by considering a set of 50 000 random samples and aggregating the corresponding function values. We consider the following aggregation methods (based on the log-scaled precision): min, mean, max, $(\max + \min)/2$. Fig. 1 illustrates the differences between these methods, for a $2d$ problem. Note that because we use log-scaled precision, the differences between instances are rather small, so we opted to only do the sampling for one instance of each BBOB problem. Based on visual interpretation of the contour plots in Fig. 1, we (somewhat subjectively) select the $(\max + \min)/2$ scaling as the most promising method.

To avoid having to constantly repeat this random sampling procedure, we also investigate the way in which the scales of the random factors, and thus the scale factors, differ across dimensions. The results are shown in Fig. 2. With exception of the smallest dimensions, the values remain quite stable. As such, we decide to implement them as hard-coded values based on the median of the shown values, rounded to the nearest decimal. The resulting factors are shown in Tab. 1.

### 3.2 Instance Creation

A second aspect to consider when combining multiple functions is the placement of the global optimum. In the previous two papers [8, 32] on affine BBOB functions, this was done based

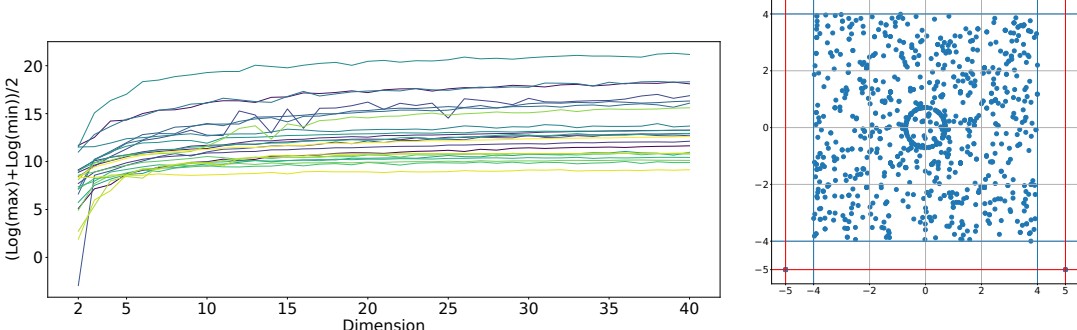

Figure 2: Evolution of the log-scaled $(\max + \min)/2$ scaling factor, rel-ative to the problem dimension. The values are based on 50 000 samples. Each line corresponds to one of the 24 BBOB functions.

Figure 3: Location of optima of the 24 2d BBOB functions. The red lines mark the commonly used box-constraints of $[-5, 5]^D$.

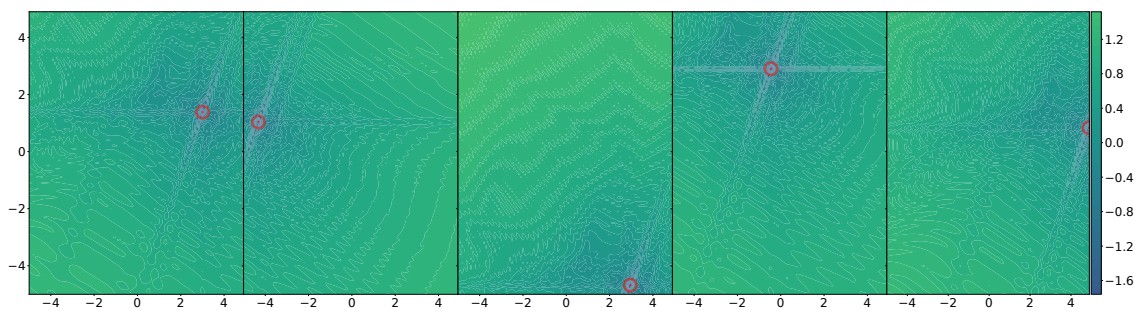

Figure 4: Log-scaled fitness values of an example of a single many-affine function with changed location of optimum.

on the instance of one of the two component functions. However, the original BBOB instance creation process can be considered somewhat biased, as not all functions make use of the same transformations [10, 18]. As such, if we extend the process of using the optimum of one of the used component functions, the optimum would be distributed as in Fig. 3. To avoid this issue, we decided to generate the optimum location separately, uniformly at random in the full domain $[-5, 5]^d$. Fig. 4 shows how a 2d-function changes when moving the optimum location.

### 3.3 Sampling random functions

As a final factor impacting the types of problems generated, we consider the way in which weights are sampled. While this can indeed be done uniformly at random (with a normalization afterwards), this might not lead to the most useful set of benchmark problems. When the weights for each function are generated this way, the probability of having a weight of 0 for any component is 0. This means that every function will contribute to some extent to the newly generated problem. As such, it would be almost impossible for this procedure to result in a unimodal problem.

One way to address this bias in function generation is to adapt how many functions are part of the newly created problem. Indeed, the combinations of two problems already lead to a vast space of interesting landscapes. We opt for a different approach: we make use of a threshold value which determines which functions contribute to the problem. The procedure for generating weights is

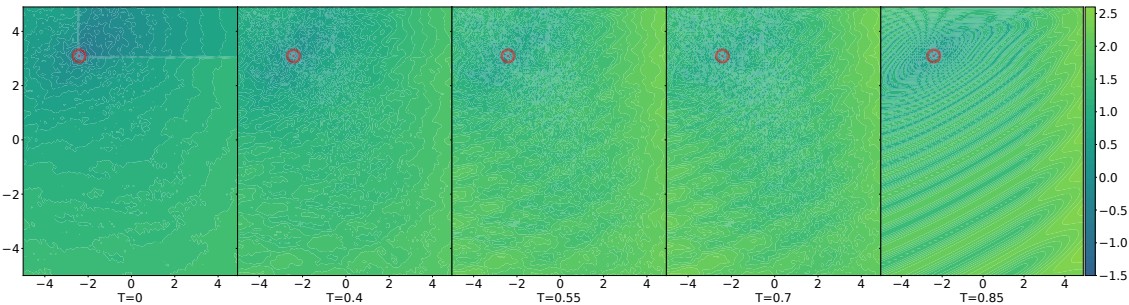

Figure 5: Log-scaled fitness values of an example of a 'single' many-affine function with 5 different sampling thresholds.

thus as follows: (1) Generate initial weight uniformly at random, (2) adapt the threshold to be the minimum of the selected value and the third-highest weight, (3) this threshold is subtracted from the weights, all negative values are set to 0. The second step is to ensure that at least two problems always contribute to the new problem. Fig. 5 provides an example of a problem generated with different threshold values. We decide to set the default value at $T = 0.85$, such that on average 3.6 problems will have a non-zero weight.

## 4 Experimental Setup

In the remainder of this paper, we will make use of 1 000 functions, with weights sampled according to Sec. 3.3 with $T = 0.85$. Each problem uses instances uniformly selected between 1 and 100 for each of the component functions, and uniformly sampled locations of the global optimum. We use the same set of weights, instances and optima locations in both 5 and 2 dimensions.

Comparing this set of generated problems with the pure BBOB functions is a key aspect of this work. To remove biases in terms of scaling, we apply the same scale factors to the BBOB functions. Practically, this means we use the all-zero weights with a 1 for the selected function to collect the BBOB data (with the location of the optima set as original). We use 5 instances of each BBOB function for our comparisons. We refer to these 'pure' BBOB functions as 'BBOB', while we refer to the MA-BBOB instances as 'affine'.

**Reproducibility**: The code used during this project, as well as all resulting data, is available at [31]. The repository also contains additional versions of the figures which could not be included here because of the page limit. We are actively working towards a data repository for MA-BBOB performance data which will also allow automated annotation via the OPTION ontology [14], for FAIR data sharing [11].

## 5 Landscape Analysis

To analyze the landscapes of the created affine problems, we make use of the pflacco package [24] to compute ELA features. We use 5 sets of $1\,000d$ points from a scrambled Sobol' sequence. We then evaluate these points and follow the advice of [25] and use min-max normalization on these function values. We finally remove all features which are constant across all problems or contain NAN values, resulting in a total of 44 remaining features. For each of these features, we then take the mean value among the 5 samples.

To gain insight into the differences between the BBOB and affine functions, we reduce the original 44 dimensional space into $2d$. To achieve this, we make use of the Uniform Manifold Approximation Projection (UMAP). To focus on the parts of the instance space covered by the newly generated problems, we create the mapping based only on the BBOB problems. The result of applying this mapping to all $2d$ problems is visualized in Fig. 6b.

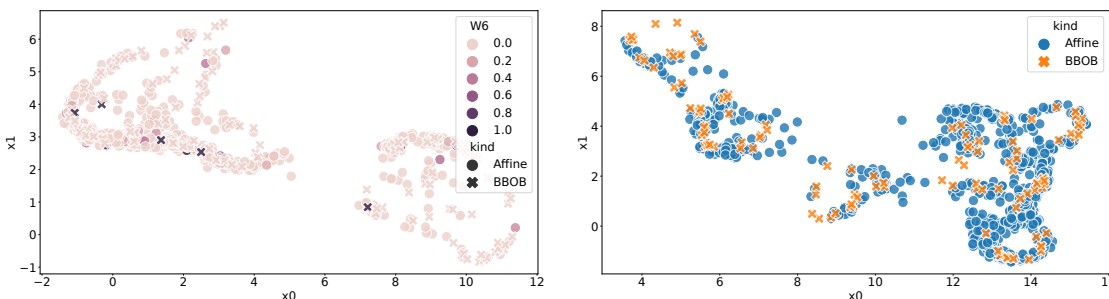

(a) Points are colored according to the weights used for BBOB function F7.

(b) Points are colored according to the function type: BBOB of affine combination.

Figure 6: UMAP-reduction of the 24 BBOB functions (5 instances each) and 1000 affine combinations for 5$d$ (a) and 2$d$ (b). The projection is created based on the BBOB only.

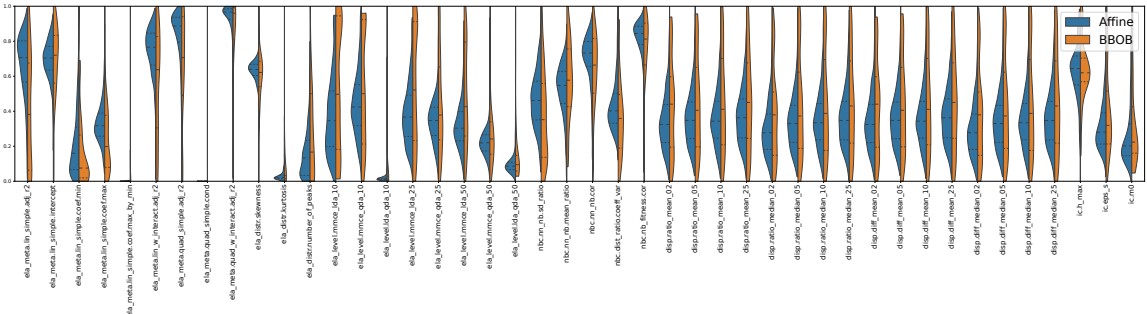

Figure 7: Distribution of (normalized) ELA feature values on the 5$d$ version of the problems.

From Fig. 6b, we observe that many of the affine problems are clustered together. While some regions between existing BBOB problems are filled, it seems that the function generation process is not able to find solutions close to every BBOB problem. This might be caused by the fact that by combining an average of 3.6 functions, it is highly unlikely that we find functions similar to e.g., a linear slope or a function with low global structure.

In addition to the dimensionality reduction, we can also investigate the distributions of individual ELA features. By comparing the distributions on the BBOB functions with the ones on the affine problems, we can gain some insight into the most common types of problems generated. In Fig. 7, we show these distributions for the min-max normalized ELA features. From this figure, we can see that for many features, the affine problems are much more clustered than the BBOB ones, which are distributed more uniformly over the space of feature values.

## 6 Algorithm Performance

While the ELA based analysis gives us some insight into the low-level characteristics of the generated problems, it does not directly give insight into the power of these problems to differentiate between algorithms. As such, we also run a set of 5 different algorithms on each problem instance. The algorithms we consider are: (1) Diagonal CMA-ES from the Nevergrad platform [26] (dCMA), (2) RCobyla from the Nevergrad platform [26] (Cobyla), (3) Differential Evolution from the Nevergrad platform [26] (DE), (4) CMA-ES from the modular CMA-ES package [6] (modCMA), and (5) L-SHADE, implemented using the modular DE package [30] (modDE).

For each of these algorithms, we perform 50 independent runs on each of the 1 000 affine functions as well as the 5 instances from each of the 24 BBOB problems. It is important to note that

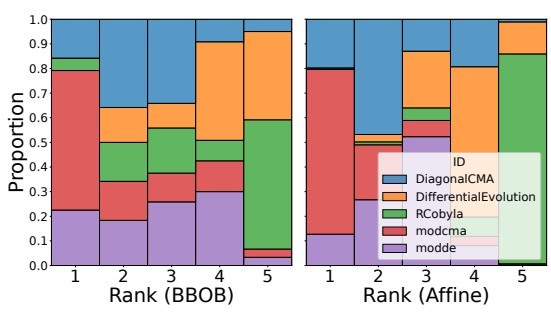
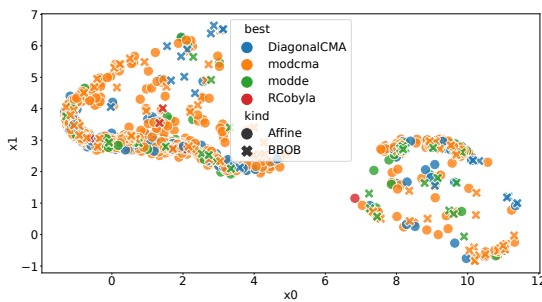

(a) Distribution of ranks based on per-function AUC after 10 000 evaluations.

(b) UMAP-reduction of BBOB functions (5 instances) and 1000 affine combinations. Projection created based on BBOB only. Color based on the algorithm with the largest AUC.

Figure 8: Results of ranking the 5 algorithms on the 5$d$ problems, based on AUC after 10 000 evaluations.

the BBOB functions make use of the same scale factors as used to generate the affine functions in order to further reduce the impact of scale differences. These experiments are performed on both the 2$d$ and 5$d$ versions of these problems.

To analyze the differences in algorithm performance between the two sets of problems, we consider the normalized area under the curve (AUC) of the empirical cumulative distribution function (ECDF) as the performance metric. For the ECDF, we use a set of 51 logarithmically spaced targets from $10^{-8}$ to $10^2$. Based on the AUC values, we then rank the set of 5 algorithms on each problem. The distribution of these ranks is shown in Fig. 8a. We observe that the overall patterns between the BBOB and affine problems are preserved. There are some notable differences, particularly with regard to the performance of Cobyla. While this algorithm often performs poorly on BBOB, for the affine problems it is ranked worst in a majority of cases. This suggests that problems where this algorithm performs well (mostly unimodal problems) are not as well-represented in the MA-BBOB functions.

In addition to this ranking, we can also link the ELA features to the algorithm performance. To explore whether the used features might correlate with the problem's difficulty from the algorithm's perspective, we link the dimensionality reduction with the best algorithm from the portfolio. This is visualized for the 5$d$ problems in Fig. 8b.

## 7 Algorithm Selection

As a final experiment, we now use the generated problems in an algorithm selection context. For each of the 5 algorithms, we train a random forest regression model to predict the AUC on each problem. The input variables for this model are either the ELA features, as is commonly done, or the weights used to generate the functions. By contrasting these approaches, we obtain an intuition for how well the ELA features capture the algorithm-relevant properties of the function.

While we can train our models in a common cross-validation manner, we can also use the same setup to test the generalizability of models trained on the original BBOB problems only. The resulting mean absolute errors MAE of these models are plotted in Fig. 9a.

We observe that the ELA representation is often worse than the weights-based one. This suggests that the used ELA features might not be sufficient to achieve generalization of an AS model. This is especially clear for the generalizability scenario, where we would have expected ELA to perform better. This poor performance seems to suggest that the ELA features might not fully capture all instance properties that determine the behavior of the algorithms.

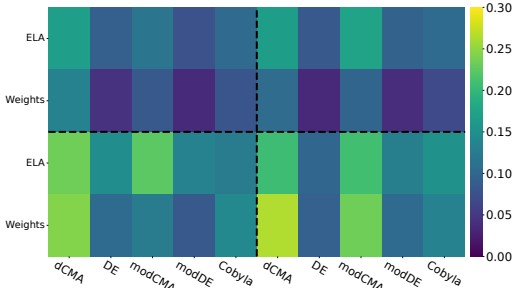
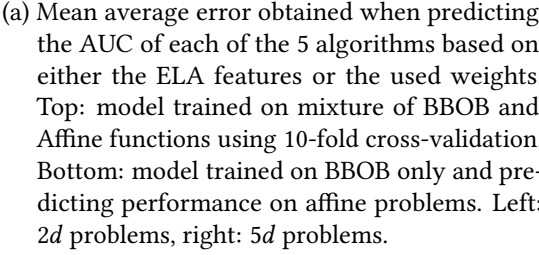
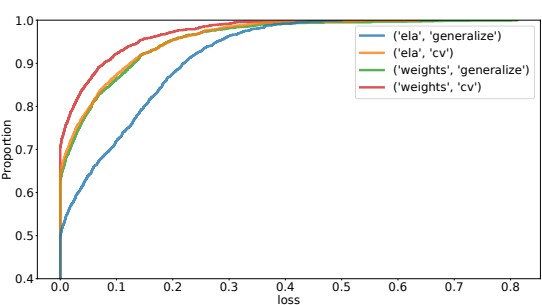

(a) Mean average error obtained when predicting the AUC of each of the 5 algorithms based on either the ELA features or the used weights. Top: model trained on mixture of BBOB and Affine functions using 10-fold cross-validation. Bottom: model trained on BBOB only and predicting performance on affine problems. Left: $2d$ problems, right: $5d$ problems.

(b) Cumulative distribution of loss (AUC) of the random forest models predicting the best algorithm ($2d$ and $5d$ problems combined), based on either the ELA features or weights-representation of the problems.

Figure 9: Performance of the random forest model predicting algorithm performance (a) or the best algorithm for each problem (b).

When training a very basic AS model (predicting the best algorithm) in the same manner (training on BBOB and evaluating on Affine), we achieve similar performance differences as suggested by Fig. 9a: the weighted F1-score based on ELA is 0.67, while the score based on weights is 0.70. The corresponding loss in terms of AUC values is plotted in Fig. 9b. This figure confirms the previous observation that the ELA features are not sufficiently representative to accurately represent the problems in a way which is relevant for ranking optimization algorithms.

## 8 Conclusions and Future Work

The proposed procedure for generating new problems as an affine combination of the 24 BBOB problems can serve as a function generator to help fill the instance space spanned by the BBOB functions. By applying a scaling step before combining the problems, we make sure that the resulting problems all have an equivalent range of objective values, regardless of the used weights. In addition, the uniform location of the global optima in the full domain avoids some of the bias of the BBOB problems. By analyzing the ELA features of 1 000 of these many-affine MA-BBOB problems, we observed that they do indeed fill a part of the instance space. There are still some inherent limitations arising from the fact that the building blocks are fixed. For example, it is impossible to generate a problem similar to the linear slope. Similarly, it is highly unlikely that new problems have specific properties such as low global structure. Nevertheless, the overall ability ranking of optimization algorithms on these problems remains similar to the ranking on the BBOB problems, suggesting that the algorithmic challenges might be similar.

The results presented above had as primary focus a first analysis of the generated MA-BBOB instances, and how they compare to the BBOB functions. For this purpose, we have considered randomly sampled instances. The selection of 'representative' instance collections still remains to be done. Another important step for future work is to test the generalization ability of AutoML systems that are trained on MA-BBOB functions and tested on numerical black-box optimization problems that do not originate from the BBOB family. In this context, our basic Random Forest-based algorithm selector indicates that the ELA features might not be as suitable for this generalization task as expected, motivating further research on feature engineering for black-box optimization.

## 9 Broader Impact Statement

After careful reflection, the authors have determined that this work presents no notable negative impacts to society or the environment.

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
