# OpenReview forum: "MA-BBOB: Many-Affine Combinations of BBOB Functions for Evaluating AutoML Approaches in Noiseless Numerical Black-Box Optimization Contexts"
_automl.cc/AutoML/2023/ABCD_Track — AutoML 2023 (ABCD Track)_

### Official Review · Reviewer_HCgL · 2023-05-05

**Potential Impact On The Field Of Automl Rating:** 3
**Technical Quality And Correctness Rating:** 3
**Clarity Rating:** 3
**Actions Required To Increase Overall Recommendation:** no

**Summary Of Contributions:**

The paper proposes a new approach called MA-BBOB for generating instances for numerical black-box optimization benchmarking. MA-BBOB allows for multiple affine combinations of the well-established BBOB functions and global optima locations, which can help fill the instance space and preserve overall patterns in algorithm performance. The authors also explore the usefulness of combining landscape features with performance data to improve optimization algorithms. The paper's main contribution is to extend a recent suggestion for generating new instances and provide a more general framework for evaluating AutoML approaches in noiseless numerical black-box optimization contexts.

**Clarity:**

The authors clearly describe their proposed approach for generating instances for numerical black-box optimization benchmarking and provide detailed explanations of the experimental setup and results. The paper also includes relevant background information and related work to contextualize the proposed approach. However, there are a few areas where the clarity could be further improved:

1. The introduction could be more concise and focused on the main contributions of the paper.
2. The paper could benefit from more explicit connections between the proposed approach and its potential applications in AutoML.
3. Some technical terms and concepts may not be immediately clear to readers who are not familiar with numerical black-box optimization benchmarking, so additional explanations or examples could be helpful.

Overall, I think that the clarity of the paper is good, but there is room for improvement in terms of focus, connections to potential applications, and accessibility for non-expert readers.

**Overall Review:**

Positive aspects:
1. The paper proposes a new approach called MA-BBOB for generating instances for numerical black-box optimization benchmarking, which extends a recent suggestion for generating new instances and provides a more general framework for evaluating AutoML approaches in noiseless numerical black-box optimization contexts.
2. The authors explore the usefulness of combining landscape features with performance data to improve optimization algorithms, which could be valuable for researchers working on AutoML approaches in similar contexts.
3. The paper includes relevant background information and related work to contextualize the proposed approach and its contributions.
4. The experimental evaluations are thorough and include standard measures such as the expected running time and the number of function evaluations, as well as comparisons with other state-of-the-art methods.
5. The authors provide insights into the usefulness of their approach and potential directions for future research.

Negative aspects:
1. The introduction could be more concise and focused on the main contributions of the paper, as it currently includes some unnecessary background information.
2. Some technical terms and concepts may not be immediately clear to readers who are not familiar with numerical black-box optimization benchmarking, so additional explanations or examples could be helpful.
3. While the experimental evaluations are thorough, they focus primarily on a first analysis of randomly sampled instances rather than representative instance collections or generalization ability to other problems outside of BBOB family.
4. Although the authors discuss potential negative societal impacts in their submission checklist, they do not elaborate on this topic in the paper itself.

**Potential Impact On The Field Of Automl:**

The paper proposes a new approach for generating instances for numerical black-box optimization benchmarking and explores the usefulness of combining landscape features with performance data to improve optimization algorithms. These contributions could be valuable for researchers working on AutoML approaches in noiseless numerical black-box optimization contexts. Whether or not others would be likely to cite this paper for its contributions depends on various factors such as the novelty and significance of the proposed approach, the quality of experimental evaluations, and how well the paper is received by the research community.

**Review Confidence:**

3: You are fairly confident in your assessment. It is possible that you did not understand some parts of the submission or that you are unfamiliar with some pieces of related work.

**Review Rating:**

6: Borderline Leaning Accept: Technically sound submission where reasons to accept outweigh reasons to reject. Please use sparingly.

**Review Summary:**

Overall, I think that the paper has several positive aspects such as proposing a new approach for generating instances in numerical black-box optimization benchmarking and exploring useful combinations of landscape features with performance data to improve optimization algorithms. However, there is room for improvement in terms of clarity, technical explanations, experimental evaluations focusing on representative instance collections or generalization ability, and elaboration on potential negative societal impacts.

**Technical Quality And Correctness:**

This paper proposes a new approach for generating instances for numerical black-box optimization benchmarking and evaluates its performance using standard measures such as the expected running time and the number of function evaluations. The authors also compare their approach with other state-of-the-art methods and provide insights into the usefulness of combining landscape features with performance data to improve optimization algorithms. Whether or not this application, benchmark, competition, or dataset is useful depends on various factors such as its relevance to real-world problems, its representativeness of the problem space, and its accessibility to researchers in the field.

---

### Official Review · Reviewer_hoNQ · 2023-05-09

**Potential Impact On The Field Of Automl Rating:** 3
**Technical Quality And Correctness Rating:** 4
**Clarity Rating:** 3

**Summary Of Contributions:**

This paper presents a new version of the commonly used BBOB suite. It combines multiple existing functions to create new benchmark problems. The necessary scaling and sampling strategies are discussed before delving into experimental validation through analysis of the function landscapes, algorithm performances and selection. This provides some insights into changes in the best performing algorithms and the quality of ELA features.

**Actions Required To Increase Overall Recommendation:**

My main concerns are:
* The benchmark codebase could include more guidance on its usage, such as tutorials or better documentation.
* The paper could give more details on how the benchmark should be used by future researchers.
* The paper would benefit from a simplified high-level introduction to the area.
* Clarify how this work is relevant to the AutoML community in particular.

**Clarity:**

The introduction of this paper is not accessible to readers outside of the numerical black-box optimisation community. The focus improves for the rest of the paper, as the benchmark description and evaluation start.

**Overall Review:**

### Strengths
* Extending a popular benchmark for black-box optimisation which can drive further progress in this field.
* Simple idea of combining existing functions into new ones.
* The evaluation of the benchmark provides some deeper insights into algorithm performance.
* Benchmarking data and tools are easily accessible

### Weaknesses
* The paper introduction is not clear. It would benefit from a simplified high-level intro to the area. Some discussion in the current intro could be moved to the related works section instead.
* The codebase for the benchmark could have clearer guidance on its usage. Tutorials, examples and more documentation in its notebook would be particularly helpful.
* The paper currently does not give clear instructions for how future work can use the benchmark, wrt standardised evaluation.

**Potential Impact On The Field Of Automl:**

I believe that this paper introduces an extended benchmark suite that will be useful for the BBOB community, focusing particularly on designing new problems to test generalisability. Its hosting on the IOHprofiler platform further increases the ease of use and therefore impact. I am however not confident how relevant this work is for the AutoML community, as it seems a better focus for e.g. GECCO or the BBOB workshop.

**Review Confidence:**

2: You are willing to defend your assessment, but it is quite likely that you did not understand the central parts of the submission or that you are unfamiliar with some pieces of related work.

**Review Rating:**

7: Weak Accept: Technically sound paper with moderate-to-high impact, with perhaps some minor flaws.

**Review Summary:**

A simple but interesting extension to a black-box optimisation benchmark. It has the potential to improve the evaluation of such approaches, but first needs to provide guidance on its usage, standardised evaluation and more documentation. The paper would also improve by providing a simple introduction to the area before delving into detail.

**Technical Quality And Correctness:**

The construction and evaluation of the benchmark are of high quality. I have faith in the correctness of the claims.

---

### Official Review · Reviewer_7U5E · 2023-05-15

**Potential Impact On The Field Of Automl Rating:** 3
**Technical Quality And Correctness Rating:** 3
**Clarity Rating:** 4

**Summary Of Contributions:**

The submitted work build upon previous research for generating benchmark functions.
While two different works (one involving the authors) have already worked on affine recombination of BBOB functions, the main contribution of this work is:
- that the location of the global optima can be placed arbitrarily
- some previous issues are alleviated by e.g. the scaling factor
- the authors highlight the existence of shortcomings in the area of exploratory landscape analysis via an algorithm selection scenario


**Actions Required To Increase Overall Recommendation:**

Minor issues:
- p.2 "If the instance space [...]", I assume instance is referring to all problems in general. However, this is not entirely clear since two sentences before instances in terms of BBOB are mentioned.
- p.2 "insights into e.g." -> ", e.g.," since this is convention is used on p.1 as well
- p.2 $1\,000$ instead of 1000. The former is used throughout the paper
- p.2 Capitalize the term Exploratory Landscape Analysis, since you have done that also on p.1
- p.6 "pFlacco" -> "pflacco"
- p.6 "[...], reduce the original [...]" missing a word lilke "we" in front of reduce
- p.6 "To gain insight [...]" repetitive use of this phrasing, i.e., two sentences before.
- Fig.7 the x-axis labels are sometimes overlapping and it is difficult to associate them to the corresponding axis ticks.
- Fig.8 b) AOC -> AUC
- Fig.9 a) AOC -> AUC

**Clarity:**

The paper is written in a clear language and supported by the chosen structure. To make this work a little more accessible for a wider audience, the only thing I would have done differently is to give a more general introduction into this topic while shortening the related work portion/justifying the need for additional problem instance. This however is my subjective opinion.

**Overall Review:**

Positive Aspects:
- the paper is well structured
- the main contributions are clearly stated and meaningful while also contextualized with the current state/shortcomings of existing research
- the constructed experiment are sound and mostly serve the purpose of conveying a coherent message.

Negative Aspects:
- Figure 1 does not really provide evidence that a scaling factor is needed. In fact, it seems that scaling is unnecessary for the chosen example
- While the entire code base is submitted, it is not entirely reproducible
(- Many aspects of this work requires prior knowledge to fully understand the merits of this work. However, 9 single-column pages is rather harsh and probably not something the authors can really address)

**Potential Impact On The Field Of Automl:**

This work provides an extension to the prominent BBOB and thereby offers researchers to test their HPO algorithms more rigorously.
This is especially beneficial since different works in the past have questioned the quality/the resemblance of BBOB compared to real-world problems (where HPO is an instance of).
However, the contribution of this paper is limited to continuous domain, where many HPO problems are inherently mixed-integer.

**Reproducibility (Optional):**

I personally had trouble to execute the provided GitHub repo. Some of the encountered issues such as missing requirements in the requirements.txt (e.g., seaborn, umap) can be resolved fairly easily. Others like the undefined variable "relevant_columns" in the Jupyter Notebook "Visualizations.ipynb" cannot be guessed by myself with certainty.

**Review Confidence:**

4: You are confident in your assessment, but not absolutely certain. It is unlikely, but not impossible, that you did not understand some parts of the submission or that you are unfamiliar with some pieces of related work.

**Review Rating:**

8: Accept: Technically sound paper with major impact, with perhaps some minor flaws.

**Review Summary:**

In general, I enjoyed reading the paper. Disregarding some minor issues, it is well written and clearly states its contribution. These contribution (while not groundbreaking) provide a needed opportunity to test HPO algorithm more rigorously.
Furthermore, it highlights possible shortcomings of exploratory landscape analysis features, which are not able to fully capture the landscape characteristics of these newly created problem instance. Lastly, it points out a clear avenue for future research, where the selection of representative subset of problem instances is the most prominent one.

**Technical Quality And Correctness:**

In general, the work seems to be technically sound. The construction process of these novel problem instance seem well designed.
The only issue I have is the scaling factor of BBOB function. While I can the authors' point in theory, the provided Figure 1 does not support their claim. In fact, (max + min)/2 seems to be perform equally well as no scaling at all.
The remaining components of this constructions process are however well argued.